# Ultraprocessed Foods and Neuropsychiatric Outcomes: Putative Mechanisms

**DOI:** 10.3390/nu17071215

**Published:** 2025-03-30

**Authors:** Mariane Lutz, Marcelo Arancibia, Javier Moran-Kneer, Marcia Manterola

**Affiliations:** 1Center for Translational Studies in Stress and Mental Health (C-ESTRES), Universidad de Valparaíso, Valparaíso 2360102, Chile; marcelo.arancibiame@uv.cl (M.A.); javier.moran@uv.cl (J.M.-K.); mmanterola@uchile.cl (M.M.); 2School of Medicine, Faculty of Medicine, Universidad de Valparaíso, Viña del Mar 2520000, Chile; 3School of Psychology, Faculty of Social Sciences, Universidad de Valparaíso, Valparaíso 2340000, Chile; 4Human Genetics Program, Institute of Biomedical Sciences, Faculty of Medicine, Universidad de Chile, Santiago 8380453, Chile

**Keywords:** ultraprocessed foods, dietary patterns, mental health, neurodegeneration, neuropsychiatric conditions

## Abstract

A body of evidence indicates an association between ultraprocessed foods (UPFs) and health outcomes. Most of it has been obtained through preclinical studies, although a number of observational studies substantiate how a high intake of these products increases the risk of neuropsychiatric disorders, and an increasing amount of dietary intervention studies confirm these findings. The aim of this narrative review is to describe some of the putative mechanisms involved in the deleterious effects of a high intake of UPFs on neuropsychiatric outcomes. A myriad of unhealthy actions may be associated with the consumption of UPFs, and some mechanisms are being discussed. They include UPFs’ high caloric density; their high sugar, sodium, and additives content and low amounts of fiber; and a high palatability that induces overconsumption, acting as obesogens. Moreover, thermal treatment of these foods generates oxidative products such as glycotoxins, lipotoxins, and acrolein, all of which affect the brain. The chemical products act, directly or indirectly, on the gut microbiome and affect the gut–brain axis, causing neuroinflammation, oxidative stress, and neurodegeneration. UPFs also exert various epigenetic effects that affect mental health and might explain the intergenerational inheritance of neuropsychiatric disorders. A diet containing a high proportion of these foods has a low nutritional density, including bioactive protective agents such as antioxidant and anti-inflammatory compounds that promote eubiosis. The evidence shows that UPFs intake affects neuropsychiatric outcomes such as neurodegeneration, cognitive decline, dementia, and mood disorders and reinforces the need to promote a healthy dietary pattern throughout all life stages, thus interfering with the current commercial determinants of health.

## 1. Introduction

As the global population is aging, mental health issues are increasing and represent a major health concern. In fact, dementia leads to an enormous burden on people and societies, and its prevalence is increasing globally [1]. An adequate diet has been identified as a modifiable risk factor for the prevention of cognitive decline [2]. On the contrary, an unhealthy dietary pattern is recognized as an important risk factor for cognitive decline, neurodegeneration, and dementia, including Alzheimer’s disease (AD) [3], the most common type of dementia. An unhealthy dietary pattern includes ultraprocessed foods (UPFs), and evidence shows associations between UPFs intake and a variety of neuropsychiatric outcomes, including age-associated neurodegeneration and dementia [4,5,6]. Moreover, a high intake of UPFs has also been associated with other mental health issues, such as mood and anxiety disorders. In this narrative review, various effects of UPFs intake on mental health are briefly described, focusing on neurodegeneration, dementia, depression, and anxiety and the possible mechanisms involved.

### Ultraprocessed Foods

NOVA, the most widely used classification system of commercial food products, is based on the extent and purpose of food processing. In this system, foods are assigned to one of the following groups: (1) contains unprocessed or minimally processed foods, i.e., the edible parts of plants or animals taken directly from nature or minimally modified/preserved; (2) contains processed culinary ingredients, such as salt, sugar, oil, or starch, produced from (1); (3) contains processed foods such as canned vegetables or freshly baked bread, produced by combining (1) and (2); and (4) contains UPFs. In this context, UPFs are formulations of ingredients, mostly of industrial use, that result from a series of processes, especially requiring sophisticated equipment and technology [7], and contain ingredients that extend palatability, profitability, and shelf-life. Examples of UPFs are carbonated soft drinks, confectionery and snacks, biscuits and cakes, mass-produced bread and breakfast cereals, reconstituted and processed meat, fish and meat alternative products, ready-to-heat and ready-to-eat meals, and desserts, among others. A prominent feature of UPFs is their high taste preference due to higher added salt, sugar, fat, and sensory-related food additives that affect texture, taste, flavor, and appearance. UPFs are generally cheaper, very attractive, convenient, and easy to prepare/consume, factors that contribute to their high intake, due to the presence of numerous additives that enhance sensory attributes [8]. Alternative classifications of UPFs are also available: a recent approach, based on machine learning, created an algorithm called the Food Processing Score, that considers not only the NOVA system but also applies this score based on grocery databases in the US to evaluate the effect of processing and the contribution of over 1000 ingredients, revealing that the price per calorie is lower when the degree of processing increases [9]. Various factors influence UPFs overconsumption, including lower socioeconomic status, lower educational status, religious preferences, low cost, and widespread availability, as well as extensive advertising, mainly affecting childhood and promoting obesity [10]. In a renowned cross-over clinical trial, Hall et al. [11] reported that the consumption of UPFs caused an increased total energy and carbohydrate intake, accompanied by rapid weight increase, compared to unprocessed food. Moreover, UPFs have been considered as obesogens that interfere with hormonal signaling, leading to adiposity [12]. Figure 1 depicts some of the UPFs effects on consumption.

UPFs characteristics include a high energy density, low nutritional density, and low supply of fiber, vitamins, and bioactive compounds with anti-oxi-inflammatory and other beneficial effects. Additionally, UPFs may contain a variety of toxins that are generated by thermal processing, such as advanced glycation end-products (AGEs) and advanced lipoxidation end-products (ALEs). On the other hand, the combination of high amounts of refined carbohydrates and fats makes them especially attractive and may exert a supra-additive effect on brain reward systems [13]. Since their food matrix has been disrupted, they are easier and faster to consume and may affect the brain more rapidly [14]. Moreover, the aggressive/persuasive marketing of UPFs is mainly oriented toward children and young adults, especially on TV and social media [15], constituting a major part of the commercial determinants of health. The overconsumption of UPFs, often triggered by stress and negative emotions, is driven in part by the effects of these products on the brain’s reward system, which amplify cravings and reduce satiety signals [16,17]. Chronic stress further exacerbates this reliance on UPFs, as individuals consume them as a coping mechanism for their immediate hedonic effects [18].

To sum up, UPFs are low cost, highly appealing, convenient, and supported by well-oriented marketing, all of which constitute barriers for approaching a healthier dietary pattern, especially when structural disadvantages are difficult to overcome, i.e., where food environments are not equitable [18,19]. Consuming large amounts of UPFs results in the replacement of foods that are the basis of a healthy food pattern. As an inexpensive source of energy, UPFs contribute to the global epidemics of obesity, hypertension, type 2 diabetes, and cardiovascular diseases, among other non-communicable diseases (NCDs) [20,21,22]. Moreover, high UPFs consumption has been associated with increased risk of all-cause mortality [23,24,25]. The intake of UPFs has risen to unprecedented levels, and the current evidence demonstrates their association with the incidence of a variety of NCDs, including neuropsychiatric outcomes [26]. On the contrary, a healthy dietary pattern is protective and favors the preservation of mental health [27].

## 2. UPFs and Mental Health—Mechanisms Involved

### 2.1. Inflammation and Oxidative Stress

UPFs consumption may be related to inflammation through mechanisms that involve a high intake of sugars (sucrose and/or high fructose syrup), salt (sodium), saturated fats, and trans fatty acids [28,29]. The induced increase in body fat is associated with low-grade inflammation, which may be explained by various mechanisms, including their low supply of antioxidants and anti-inflammatory compounds, prebiotics or fermentable fiber, and their low nutritional density (vitamins and minerals), along with the high intake of sugar and refined carbohydrates (starch), low quality fats, and numerous additives, among others. Moreover, plastic food packaging may also be a source of chemicals, including bisphenol and phthalates, that act as endocrine disruptors, increasing the levels of inflammatory biomarkers such as CRP, IL-6, and IL-10, contributing to the development of NCDs [30]. Several cross-sectional studies have reported higher levels of these substances in the urine of people with high UPFs consumption [31,32,33,34,35].

In the central nervous system (CNS), microglia are essential for maintaining normal function by exerting both neuroprotective and neurotoxic effects. Microglia activation by inflammatory mediators and cytotoxic molecules, i.e., lipopolysaccharide (LPS), interferon-γ (IFN-γ), β-amyloid (Aβ), and α-synuclein (α-syn), generate neuroinflammation, reactive oxygen species (ROS), and nitric oxide (NO), exacerbating neurotoxicity, which is pivotal in AD and other dementias [36,37]. Preclinical studies show that UPFs components (i.e., nanosized particles contained in additives, trans fatty acids, and bisphenol A) can disrupt the amygdala–hippocampal complex, a key region for emotion regulation [38,39]. Evidence associates the high content of additives in UPFs with oxi-inflammation. In fact, inflammatory processes have been implicated in the pathophysiology of depression, and the dysregulation of both the innate and adaptive immune systems occurs in depressed patients and hinders favorable prognosis, including antidepressant responses [40].

Oxidative stress involves the generation of ROS that lead to structural modifications of cellular functioning [41]. In lipids, oxidation is paramount in polyunsaturated fatty acids (PUFAs) through chain reactions that generate products that damage cells. Oxidative stress may induce structural and physiological damage in the CNS [42], due to a deficit in brain antioxidants, since the blood–brain barrier (BBB) reduces the diffusion of some protective agents and since cell damage is not easily repaired due to the limited capacity of neuronal regeneration [43]. CNS structures contain high amounts of long-chain PUFAs, in particular docosahexaenoic acid (DHA, 22:6*n*-3), arachidonic acid (AA, 20:4*n*-6), and eicosapentaenoic acid (EPA, 20:5*n*-3), all of which are relevant biomolecules [44] very susceptible to oxidation.

### 2.2. Thermal Treatment of Foods and Generation of Toxins

#### 2.2.1. AGEs and ALEs

AGEs and ALEs are renowned sources of toxins in UPFs. These compounds are formed when foods are submitted to high temperature, mainly in a dry environment. AGEs are glycotoxins formed as a final product of the Maillard reaction, involving reducing sugars and proteins, lipids, and/or nucleic acids [45]. Through the promotion of oxidative stress, AGEs induce the activation of transcription factors, with the production of inflammatory mediators such as cytokines and acute-phase proteins [46]. In fact, the generation of AGEs is often accompanied by oxidative reactions involving ROS. Moreover, the union of AGEs to specific receptors for these compounds (RAGEs) is considered as a critical signaling pathway responsible for activating the genes linked to the inflammatory responses involved in AD [47,48]. Studies in mice fed a diet containing less AGEs throughout their life showed a reduction in systemic oxidative stress, metabolic tissues, and organs and a longer life expectancy vs. animals consuming higher AGEs amounts [49]. Various animal models and epidemiological studies have shown a connection between AGEs and cognition [50]. In general terms, animal studies report adverse effects involving altered gut microbiota (GM) composition, damage to colonic epithelia, neuroinflammation, and cognitive impairment [51]. In a cross-sectional study, Deng et al. [52] analyzed baseline data from the Chinese Healthy Dance Study, measuring plasma levels of various AGEs, and applied cognitive tests that measure different cognitive domain scores, concluding that higher AGE levels were negatively correlated with memory, attention, and executive and language functions. Moreover, higher plasma AGEs levels were associated with higher odds of mild cognitive impairment.

AGEs promote oxidative stress in the brain, which may lead to their formation as well [53]. Methylglyoxal, a biomarker of AGEs, increases endogenous ROS formation in brain cells, affecting BBB integrity and damaging or killing microglia/brain immune cells [54]. Additionally, oxidative protein damage accelerates the formation of oligomers that aggregate in neurons, in normal aging brains and age-related neurodegenerative disorders, such as AD and Parkinson’s disease [55]. Oxidative stress also induces the formation of highly reactive compounds causing covalent modifications of proteins that result in the generation of ALEs, which exert various effects, ranging from protein, DNA, and phospholipid damage to signaling pathway activation and/or alteration [56]. The generation of reactive ALEs through lipid peroxidation (and glycoxidation) reactions generate protein cross-linking, oligomerization, aggregation, and formation of protein oxidation adducts that are involved in aging and neurodegenerative diseases [57]. Consequently, the AGE/ALE pool represents a biomarker for the risk of age-related and neurodegenerative diseases, and efforts can be made to decrease their intake through a healthy dietary pattern, including the reduction in the intake of UPFs [58].

#### 2.2.2. Acrolein and Acrylamide

The high-temperature treatment of foods may induce the formation of acrolein, a highly reactive aldehyde formed as a byproduct of the pyrolysis of carbohydrates and lipoxidation, forming acrylamide, i.e., in fried foods [59,60]. Acrolein promotes oxidative damage, which may induce neurodegenerative diseases [61,62,63]. It depletes endogenous antioxidants (i.e., glutathione), generating ROS, promoting oxidative stress, thus damaging proteins and DNA; affecting phospholipids; and inducing neurodegenerative diseases [64]. Significant amounts of acrolein have been observed in the brain of AD patients [65]. Huang et al. [31] reported the finding of acrolein in the brain and spinal cord in patients suffering from neurodegenerative diseases in which polyamine synthesis and metabolism are increased, which leads to the production of hydrogen peroxide and aldehydes associated with cell death [66]. The neuropathology hallmarks of AD are senile plaques and neurofibrillary tangles, deficits in axonal transport, synaptic dysfunction, and neuronal loss. The association of acrolein with AD involves the induction of hyperphosphorylation of microtubule-associated protein tau [67] and the promotion of Aβ aggregation in senile plaques [68], contributing to the pathogenesis of the disease [69]. 

### 2.3. Gut Microbiota

The amount and composition of the GM are affected by various factors, including gender, age, diet, and geographical zone. The microbial–gut–brain axis is a complex network of connections that involves the nervous, endocrine, and immune systems via multiple pathways (metabolic, immune, neurotransmission, neuroendocrine–hypothalamic axis, gut, and BBB), which may influence the onset and progression of neurodegenerative diseases. An abnormal balance in GM (or dysbiosis) is associated with various neuropsychiatric disorders [70], including autism spectrum disorder [71], depressive disorders (DD) [72], Parkinson’s disease [73], and AD [74]. In fact, communication between GM and the brain has generated a great deal of research on the issue of gut dysbiosis and its association with alterations in the CNS as a trigger for neurodegenerative diseases [75].

GM metabolites include tryptophan, γ-aminobutyric acid (GABA), histamine, serotonin, short-chain fatty acids (SCFAs), dopamine, and acetylcholine, all of which are involved in the regulation of brain activity [76]. Erny et al. [77] described how a reduced biomass and the complexity of the GM impairs the brain’s macrophages (microglia), and how the presence of SCFAs (acetic, propionic, and butyric), produced by bacterial fermentation of prebiotic fiber, restores the maturation, morphology, and function of microglia, which highlights the importance of a well-balanced diet on the maintenance of adequate bidirectional GM–brain communication. SCFAs are neuromodulators: they inhibit neuroinflammation and regulate the enteric neuroendocrine system; protect the integrity of the BBB, neuroplasticity, brain function, and behavior; and modulate the synthesis of neurotransmitters and their receptor’s expression [78,79]. Moreover, persistent systemic inflammation, associated with a low-fiber diet, can affect the BBB structure, increasing permeability and precipitating neuroinflammation, neurodegeneration, and age-related cerebral changes [80]. Inadequate GM is a source of amyloid proteins produced by various bacterial strains (i.e., *Escherichia coli*, *Bacteroides fragilis*, and *Salmonella typhi*), causing misfolding of Aβ oligomers and fibrils that may compromise the immune system [81]. A dietary pattern with a high content of UPFs leads to dysbiosis and, consequently, may trigger the development of neurodegenerative diseases [82,83]. Figure 2 summarizes these findings.

### 2.4. Epigenetic Mechanisms: Implications for Gene Regulation and Inheritance

UPFs consumption has been increasingly implicated in epigenetic modifications that may contribute to the development of neuropsychiatric disorders. These epigenetic alterations result from a complex interplay of mechanisms—such as nutritional deficiencies, oxi-inflammation, and GM alterations—that converge to modify chromatin condensation. Such modifications can alter the expression of genes essential for neuronal function and plasticity, thereby influencing neuropsychiatric outcomes [84]. Mechanistically, UPFs influence the epigenome of cells by mainly three pathways: (1) This can occur by inducing abnormal DNA methylation patterns, a key epigenetic mechanism that involves the addition of methyl groups to cytosine residues within CpG islands, thereby influencing gene expression, i.e., excessive sugar intake has been associated with hypermethylation of genes involved in metabolic regulation [85,86]. Hypermethylation can silence genes essential for insulin signaling and lipid metabolism, leading to metabolic dysregulation [86]. Moreover, methylation changes can be inherited by offspring, potentially predisposing them to similar metabolic disorders [85]. (2) The high-fat and sugar contents of UPFs can disrupt normal histone acetylation patterns, leading to either an open or condensed chromatin state and an altered expression of genes related to inflammation and metabolic processes [87]. (3) This can occur by modulating the expression of ncRNAs involved in metabolic and inflammatory pathways. Studies have demonstrated that dietary fatty acids and food additives can influence the expression of miRNAs that regulate genes implicated in lipid metabolism and insulin signaling [87,88,89]. Furthermore, alterations in miRNA profiles can be transmitted to offspring through germ cells, potentially perpetuating disease risks across generations [90,91]. UPFs can also modulate epigenetic marks via AGEs, which influence the methylation of the RAGEs promoter and the expression of RAGEs in cells [92]. This effect is particularly pronounced under hyperglycemic conditions, inducing increased *H3K4* methylation and reduced *H3K9* methylation on inflammatory genes such as NF-κB [93]. Most studies investigating the link between diet and epigenomic changes have primarily focused on analyzing CpG regions associated with metabolic outcomes. However, some findings have also revealed alterations in the methylation patterns of regions related to biological pathways involved in brain processes and neuropsychiatric outcomes.

Diets high in UPFs can also lead to nutritional deficiencies, as well as the intake of additives and contaminants such as heavy metals, and epigenetic factors probably linked to autism and attention deficit/hyperactivity disorder. These diets may cause dysfunction in metallothionein genes and suppression of the paraoxonase-1 gene (*PON1*), both vital for detoxifying heavy metals and protecting neural pathways [94], highlighting the importance of addressing healthy dietary patterns. Interestingly, a meta-analysis of epigenome-wide association studies (EWASs) showed that UPFs affect the epigenome of children by changing the DNA methylation pattern in seven CpG sites [95]. Among these, three sites were hypomethylated—one mapped to the autism-related and neuropsychiatric disorder-related gene *PHYHIP*, while the other two were located in *NHEJ1* and *ATF7* genes, associated with neurodevelopmental disorders and behavioral regulations [96]. Thus, UPFs consumption is linked to potential neuropsychiatric outcomes in children through epigenetic mechanisms in key genes. Another EWAS meta-analysis exploring the epigenomic effects of diets with a high glycemic index/load showed a positive correlation between glycemic load and the methylation of a CpG site within the *WDR27* gene—which plays a role in cellular scaffolding for protein interactions [96]. Domínguez-Barragán et al. [97] identified 18 differentially methylated CpG sites associated with overall diet quality through an EWAS. Among them, *MAST4* and *AHRR* were highlighted as having potential roles in neuropsychiatric phenotypes. Lower methylation levels of *MAST4*, which were associated with low-quality diets, have been connected via GWAS [98] to reduced hippocampal volume—a structural change consistently observed in depression [99]. Conversely, higher *AHRR* methylation, associated with high-quality diets, may influence neuropsychiatric outcomes through its regulation of the aryl hydrocarbon receptor, expressed in neurons, oligodendrocytes, and endothelial cells, which is also emerging as a key regulator of the gut–brain axis [100]. Collectively, these studies underscore the potential for UPFs consumption to modulate epigenetic mechanisms that may contribute to neuropsychiatric outcomes, highlighting an intricate relationship between nutritional exposures, gene regulation, and brain health.

Research shows that UPFs intake can alter the epigenetic profile of sperm by changing DNA methylation patterns on key genes, potentially impacting fertility and the developmental health of future generations [94,101]. Additionally, interactions between the host and GM in response to UPFs can leave lasting epigenetic marks on germ cells [102], influencing inherited traits. This raises the question of whether the epigenetic changes seen in brain and nervous system genes from UPFs consumption are also present in germ cells, which could help explain the intergenerational transmission of neuropsychiatric disorders. Overall, these findings underscore the potential for dietary interventions to reduce the risk of mental health and neurodegenerative conditions, highlighting the importance of public health policies aimed at curbing UPFs consumption.

## 3. Mental Health-Related Outcomes

Most studies on the association between UPFs intake and psychiatric outcomes are observational, in which UPFs intake has been consistently associated with a series of mental health issues, although the most related mental health outcome is DD. Most human studies describing the association between UPFs intake and cognitive decline are observational, mainly longitudinal. Li et al. [6], in a prospective cohort study using the UK Biobank data, reported that a higher intake of UPFs was associated with a higher risk of dementia, whereas substituting unprocessed or minimally processed foods for UPFs was associated with a lower risk of dementia. Gomes et al. [68], in a prospective study developed in Brazil, evaluated changes in cognitive performance over time through word recall, word recognition, phonemic and semantic verbal fluency tests, and the Trail-Making Test B version, during a median follow-up of 8 years, reporting that a high percentage of daily energy consumption of UPFs was associated with cognitive decline in adults. These and other observational studies reflect the existence of a variety of methodological differences regarding demography (age, sex, and race), storage of fecal samples (i.e., to analyze GM), analytical methods, self-reported screening questionnaires to diagnose depression/anxiety, assessment of dietary intake, and reporting of study findings, as well as the use of psychotropics, all of which, beyond the dietary pattern, can influence their results.

In a prospective French study, Adjibade et al. [103] reported that participants in the highest quartile of UPFs intake had a 31% increased risk of developing DD, concluding that for every 10% increase in UPFs consumption there was a 21% increased risk of depression. Similarly, Gomez-Donoso et al. [104] followed Spanish university graduates without depression for 10.3 years, and participants in the highest quartile intake of UPFs had a 33% higher risk of developing DD than those in the lowest quartile. In a cross-sectional study, Hecht et al. [105] explored adverse mental health symptoms and mentally unhealthy days in a representative sample of the US population, observing that those with the highest level of UPFs intake were more likely to report at least mild depression, were more mentally unhealthy, had more anxious days per month, and were less likely to report zero mentally unhealthy or anxious days. In a systematic review and meta-analysis of observational studies, Mazloomi et al. [106] concluded that UPFs consumption is related to an increase in the risk of depression. Similarly, Lane et al. [107] analyzed the association between UPFs consumption and various mental symptoms and disorders through 15 cross-sectional and 2 cohort studies, reporting that UPFs intake was associated with increased risk of subsequent depression. In the umbrella review and meta-analyses of observational evidence of Dai et al. [108], the authors classify the associations between UPFs consumption and type 2 diabetes, obesity/overweight, depression, and common mental disorders as ‘highly suggestive’, mainly due to the heterogeneity of the methodological considerations. From prospective evidence, a greater UPFs intake was associated with an increased risk of subsequent DD. A series of meta-analyses of cross-sectional studies showed that higher UPFs consumption was associated with common mental disorder, depressive, and anxious symptoms. These preliminary findings point to potential links between UPFs and psychiatric outcomes, mainly depressive/stress-related disorders, nonetheless the degree of heterogeneity of the studies. Longitudinal studies appear to reinforce these associations, providing additional insights. For instance, Arshad et al. [109] investigated the recurrence of DD symptoms (two or more episodes) in British participants over a 13-year period, reporting that those in the highest quintile of long-term UPFs intake had a 31% higher likelihood of recurrent depressive symptoms compared to those in the lowest quintile. The study identified a dietary pattern among high UPFs consumers characterized by excessive intakes of sugar, sodium, saturated and trans-fatty acids, and additives (e.g., emulsifiers and preservatives), combined with lower consumption of protective foods like fruits, vegetables, and fish. Similarly, Leal et al. [110] analyzed longitudinal data in Brazil, observing that participants in the highest quartile of UPFs intake had a significantly higher risk of developing DD, with the main UPFs consumed being bonbons, soft drinks, sliced white bread, hot dogs/hamburgers, and margarine. More recently, Ghernati et al. [111] explored the relationship between dietary quality and mental health in a five-year study of Lebanese adults, finding that higher UPFs intake was associated with increased odds of depression and anxiety symptoms. These findings are summarized in Table 1 and align with broader evidence, suggesting that the hyperpalatability of UPFs, coupled with their high energy density and low nutritional quality, promote dysregulated eating behaviors such as overeating and emotional eating and neuropsychiatric outcomes.

Additionally, a series of systematic reviews and a meta-analysis describes an association between UPFs-induced dysbiosis and DD. Simpson et al. [112] analyzed the associations between GM, anxiety, and depression in 26 studies, the quality of which was rated as ‘Good’ for only 3, considering their risk of bias, mainly due to their heterogeneity, including factors such as sex, medications, diet, and methodologies. In spite of these considerations, which are common in these studies, anxiety and DD may be characterized by a higher relative abundance of proinflammatory species, and a lower abundance of SCFA-producing bacteria, which is a feature of the intake of UPFs. Radjabzadeh et al. [113] reported the effect of GM diversity and composition on depression scores, using Mendelian randomization, reporting associations between different taxa with depressive symptoms, while Gao et al. [114] compared the GM between subjects with DD and healthy controls, in a systematic review and meta-analysis, in which there were no significant differences on alpha diversity indices, *Firmicutes,* and *Bacteroidetes* between both groups. However, by further studying subgroups, the authors observed depleted or higher levels of a series of other species in patients with DD, additionally observing an effect of the use of psychotropic medication. Additionally, Brushett et al. [115] reported differences in GM between patients with DD according to the use/not use of psychotropics. These and other observational studies reiterate the importance of a reduction in the anti-inflammatory SCFA butyrate-producing bacteria on DD. Proinflammatory genera prevail in subjects with DD, in association with high UPF intake, supporting that the modulation of the inflammatory process, e.g., changing the diet and/or consuming probiotics, may have therapeutic benefits for patients with DD [116,117,118,119]. In this context, dietary psychobiotics may play major roles in improving GM, thus impacting mental health, and are increasingly being considered as an alternative in the treatment of DD [120].

A summary of the putative mechanisms involved in the effects of UPFs on neuropsychiatric outcomes is depicted in Figure 3.

## 4. Discussion

A number of studies indicate that the intake of UPFs is associated with a myriad of deleterious health effects, including neuropsychiatric mental health outcomes such as neurodegeneration, cognitive decline, dementia, and mood disorders. UPFs consumption is determined by various factors, including socioeconomic and cultural determinants, their availability and aggressive marketing, their chemical composition (high energy density, low nutritional density, and low fiber content), the effects of processing (generation of toxins, use of additives, and types of packaging), and the indirect effects on food intake (obesogens and overconsumption). Although preclinical studies demonstrate many of the mechanisms involved in the harmful consequences of a high UPFs intake, most of the evidence available on their putative effects on mental health outcomes has been obtained from observational studies, while dietary intervention (clinical) studies are still not enough to determine causality. In spite of the evidence of the detrimental effects of UPFs growing and being well recognized, public policies oriented to the reduction in their intake are scarce. As the population ages, mental health issues are more prevalent, and efforts should be made to prevent them through better lifestyles, i.e., maintaining healthy food patterns throughout one’s lifespan. Reducing the intake of UPFs by interfering with the current commercial determinants of health (namely, food intake) may be considered as a major step toward mentally healthy aging.

## Figures and Tables

**Figure 1 nutrients-17-01215-f001:**
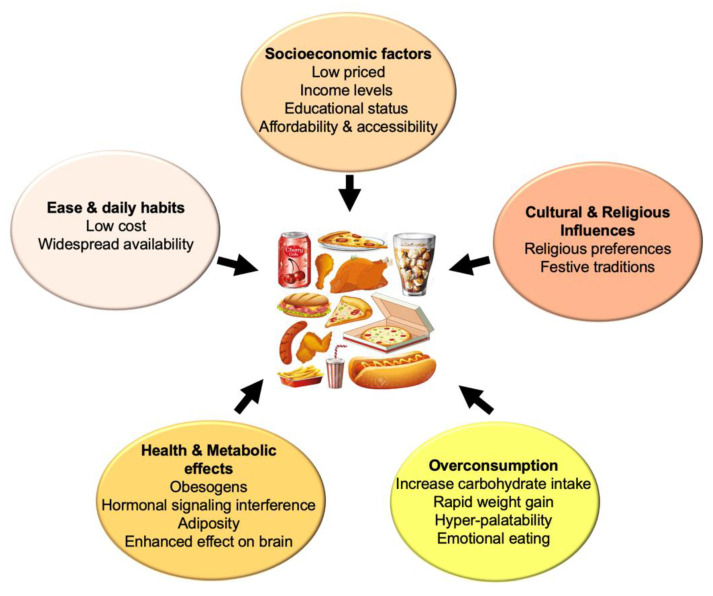
Socioeconomic, lifestyle, and metabolic factors influencing UPFs consumption.

**Figure 2 nutrients-17-01215-f002:**
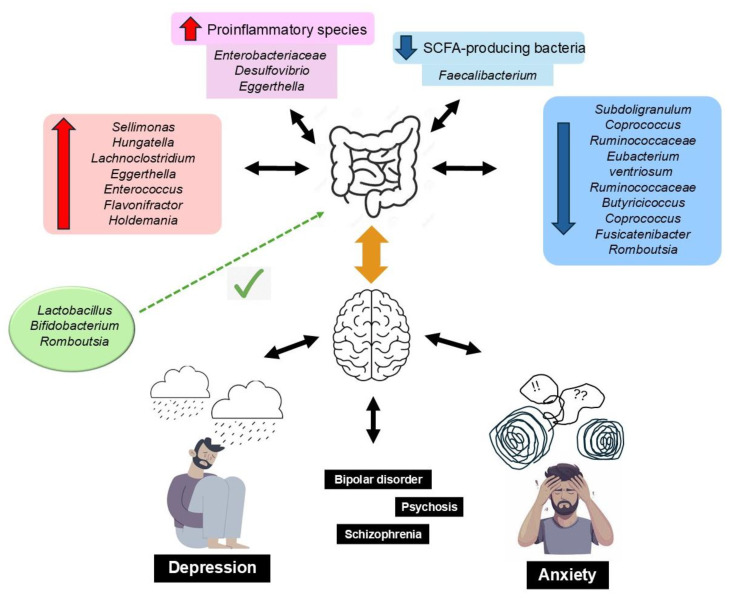
Dysbiosis of the gut microbiota is associated with anxiety and depressive disorders, resulting from alterations in the gut–brain axis. This is characterized by an increase in proinflammatory bacteria and a decrease in SCFA-producing bacteria. This imbalance is reflected in reduced microbial richness and altered diversity, with specific taxa linked to the severity of depressive symptoms across various internalizing disorders (e.g., major depressive disorder, bipolar disorder, psychosis, and schizophrenia). Conversely, higher levels of *Lactobacillus*, *Bifidobacterium*, and *Romboutsia* have been identified as potentially protective against neuropsychiatric conditions.

**Figure 3 nutrients-17-01215-f003:**
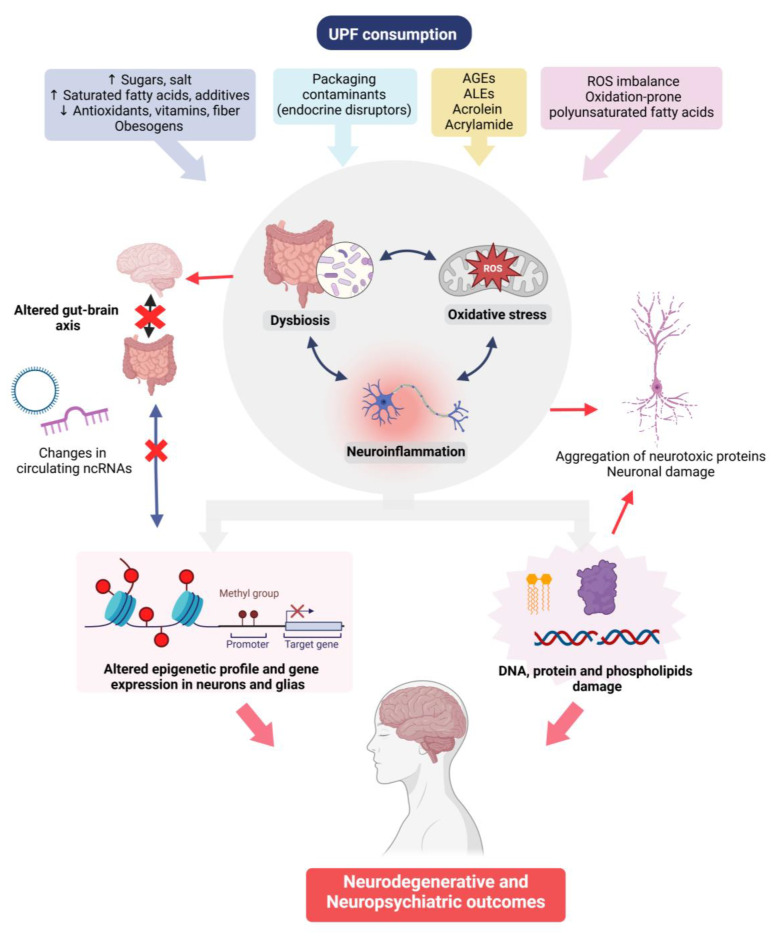
Possible mechanisms involved in the association between UPFs intake and neuropsychiatric health outcomes. AGEs: advanced glycation end-products; ALEs: advanced lipoxidation end-products; ROS: reactive oxygen species; and ncRNA: non-coding RNA. Created in BioRender. Manterola, M. (2025) https://BioRender.com/pswx20q (accessed on 3 March 2025).

**Table 1 nutrients-17-01215-t001:** Examples of observational studies showing significant associations between UPFs intake and anxiety/depressive outcomes.

Results and Conclusions	Type of Study	N° Studies and Subjects (N)	Reference
↑ intake of UPFs correlates with ↑ risk of depressive symptoms, as indicated by 2221 incident cases.	Longitudinal study, 5.4 years	N = 26,730 (20,380 women, 6350 men) without depressive symptoms	Adjibade et al., 2019 [103]
Individuals with the highest consumption of UPFs exhibited ↑ risk of depression, as demonstrated by 774 incident cases.	Longitudinal study, 10.3 years	N = 14,907 free of depression	Gómez-Donoso et al., 2019 [104]
↑ consumption of UPFs is associated with mild depression, anxiety, and mental unhealth.	Cross-sectional study, 5 years	N = 10,359	Hecht et al., 2022 [105]
UPFs are linked to ↑ risk of depression, with no similar association observed for anxiety. Notably, for every 10% ↑ in UPFs intake as a share of daily calories, there is an 11% ↑ in depression risk, indicating a dose–response relationship.	Systematic review and meta-analysis	26 studies N = 260,385	Mazloomi et al., 2023 [106]
↑ consumption of UPFs is associated with ↑ odds of depressive and anxiety symptoms, and ↑ intake of these foods ↑ the risk of developing depression over time.	Systematic review and meta-analysis (15 cross-sectional, 2 prospective)	17 studies N = 385,541	Lane et al., 2022 [107]
UPFs consumption is associated with depression and 25 additional adverse health outcomes.	Umbrella review, meta-analysis	39 studies, of which 4 were on depression	Dai et al., 2024 [108]
British people with high UPFs intake associate with recurrent depressive symptoms and has ↑ odds of experiencing these symptoms compared to those with low UPFs consumption.	Longitudinal study, 13 years	N = 4554	Arshad et al., 2024 [109]
Participants in the highest quartile of UPFs consumption had ↑ risk of developing depressive disorders.	Longitudinal study, 2.96 years	N = 2572	Leal et al., 2023 [110]
Increasing UPFs intake is associated with ↑ odds of depression.	Cross-sectional study	N = 188	Ghernati et al., 2025 [111]

↑: high/higher.

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
