# Peer review of "Ultraprocessed Foods and Neuropsychiatric Outcomes: Putative Mechanisms"

_nutrients, 2025, doi:10.3390/nu17071215_

Round 1
Reviewer 1 Report
Comments and Suggestions for Authors
To the Authors
The review (Ms ID nutrients-3538782) dwelves on a topic of large interest in the nutrition field (UPF and NPD) and a current hot one. Current understanding on the negative effects of UPF on health and NPD is building up against the obvious growing global commercial/marketing interests in promoting and selling UPF. Unfortunately, as already clearly underlined by the Authors, very little cause-effect demonstrations are available in the human/clinical field despite a large body of knowledge at their experimental level. Even more difficult is to actually envisage practical solutions at the population-scale level.
Major Point
Overall, although the review is relatively well designed and sufficiently well written, the present review is adding relatively little insights to the field.
Minor Points
- It would helpful to clearly separate cause-effect demonstrations from statistical associations;
- I would suggest to include some summarizing table of the current knowledge;
- Title: I would suggest to slightly modify the title from “Possible associations between Ultraprocessed Foods and Neuropsychiatric Outcomes: Putative mechanisms” to “Ultraprocessed Foods and Neuropsychiatric Outcomes: Putative mechanisms”
- Abstract: LL 12-13 (A body of evidence demonstrates an association between ultra-processed foods (UPFs) and health outcomes). I would suggest to change the verb “demonstrates” into “indicates”.
- Text; I would suggest polishing the grammar and style in order to achieve a more concise and impactful information and concept flow. The text size could safely be consistently shortened up.
- Introduction. This section is clear enough although some statements either need a reference quotation or should be removed;
- Section 1.1: I would suggest concentrating on UPF (please consider including some summarizing table) and deserve the anticipation of their negative effects on subsequent paragraphs. Several sentences appear to contain redundant information
- Section 2.1: The paragraph on the relationships between UPF and inflammation could be improved. The paragraphs, as it is, is conveying little insights beyond a list of well-known knowledge;
- Section 2.2: It is to be reminded that oxidative stress is a component of inflammation;
- Section 2.4: The Authors are often reporting the analyses of other Authors without a clear personal take or expert opinion on the issue. This aspect should be improved;
- Conclusion. I would suggest shortening this section by focusing on key conclusions only.
- References: The bibliography is sufficiently updated and relevant although I would suggest strictly avoiding citing review papers.
Author Response
Major Point
Overall, although the review is relatively well designed and sufficiently well written, the present review is adding relatively little insights to the field.
Minor Points
- It would helpful to clearly separate cause-effect demonstrations from statistical associations;
- I would suggest to include some summarizing table of the current knowledge;
- Title: I would suggest to slightly modify the title from “Possible associations between Ultraprocessed Foods and Neuropsychiatric Outcomes: Putative mechanisms” to “Ultraprocessed Foods and Neuropsychiatric Outcomes: Putative mechanisms”
- Abstract: LL 12-13 (A body of evidence demonstrates an association between ultra-processed foods (UPFs) and health outcomes). I would suggest to change the verb “demonstrates” into “indicates”.
- Text; I would suggest polishing the grammar and style in order to achieve a more concise and impactful information and concept flow. The text size could safely be consistently shortened up.
- Introduction. This section is clear enough although some statements either need a reference quotation or should be removed;
- Section 1.1: I would suggest concentrating on UPF (please consider including some summarizing table) and deserve the anticipation of their negative effects on subsequent paragraphs. Several sentences appear to contain redundant information
- Section 2.1: The paragraph on the relationships between UPF and inflammation could be improved. The paragraphs, as it is, is conveying little insights beyond a list of well-known knowledge;
- Section 2.2: It is to be reminded that oxidative stress is a component of inflammation;
- Section 2.4: The Authors are often reporting the analyses of other Authors without a clear personal take or expert opinion on the issue. This aspect should be improved;
- Conclusion. I would suggest shortening this section by focusing on key conclusions only.
- References: The bibliography is sufficiently updated and relevant although I would suggest strictly avoiding citing review papers.
Answer: the suggestions were considered, and all changes are written in red. A new figure and a new table were added, among other changes according to the recommendations, as described in the Cover Letter for version 2

Reviewer 2 Report
Comments and Suggestions for Authors
Title:
Possible associations between Ultraprocessed Foods and Neuropsychiatric Outcomes: Putative mechanisms
The manuscript by Lutz M. et al. is a review that collects and discusses data on the effects of UPFs intake on mental health, and the potential mechanisms involved.
It is an interesting article that adds new knowledge to this field of research about unhealthy actions with an impact on mental health, in the current context of an increased consumption of UPFs.
The information collected has been organized and detailed in the introduction and 2 other sections (sections 2 (subsections 2.1.-2.5.) and 3), and in 2 figures, all supported by a body of evidence of 123 relevant references, of which a significant number are recent publications, from the last 5 years.
The manuscript is clear and well written; however, a few minor points should be modified before publication.
General comments:
Please be consistent with all abbreviations in the manuscript, including their meanings the first time they are mentioned (e.g.: CRP, miRNAs, ncRNAs, etc.).
If possible, reduce the use of rarely used abbreviations: for example:
CVD, DHA, EPA, GABA, 5-HT - appear only 1 time in the manuscript;
T2D - 2 times.
Specific comments:
Introduction and Body of text:
OK
Figures are appropriate and easy to understand/interpret.
Conclusions:
These coincide with the review's findings and have been summarized correctly.
Please change the numbering of the Conclusions section (line 433): with 4 instead of 5.
References:
The references are appropriate.
The self-citation rate is low (<2%).
Final conclusions:
In my opinion, the authors have produced a comprehensive work of great interest to the journal's readers.
The topic is of great relevance.
I therefore suggest minor revisions to improve its quality before publication.
Author Response
Please be consistent with all abbreviations in the manuscript, including their meanings the first time they are mentioned (e.g.: CRP, miRNAs, ncRNAs, etc.). If possible, reduce the use of rarely used abbreviations: for example: CVD, DHA, EPA, GABA, 5-HT - appear only 1 time in the manuscript; T2D - 2 times.
Specific comments:
Introduction and Body of text: OK
Figures are appropriate and easy to understand/interpret.
Conclusions: These coincide with the review's findings and have been summarized correctly. Please change the numbering of the Conclusions section (line 433): with 4 instead of 5.
References: The references are appropriate. The self-citation rate is low (<2%).
Final conclusions: In my opinion, the authors have produced a comprehensive work of great interest to the journal's readers. The topic is of great relevance.
Answer: suggestions were considered, and changes are written in red. A new figure and a new table were added, among other changes according to the recommendations. Please see details in the Cover Letter file

Reviewer 3 Report
Comments and Suggestions for Authors
Reviewer Report
Manuscript Title: Possible associations between Ultraprocessed Foods and Neuropsychiatric Outcomes: Putative mechanisms
The manuscript addresses a timely and significant topic concerning the impact of ultra-processed food (UPF) on neuropsychiatric health. Increasingly, research indicates the negative health effects associated with UPF consumption, as well as the benefits of consuming less processed foods. The authors thoroughly discuss potential mechanisms of UPF influence, covering both biological aspects and socio-economic determinants, which adds great value for readers interested in this issue.
The manuscript is based on an extensive literature review, including both epidemiological and experimental studies, which enhances its credibility. The emphasis on prevention and the need for public health policies to reduce UPF consumption is particularly important. However, there are aspects that require refinement to improve the manuscript's clarity and scientific value.
My Comments:
- Complexity of Mechanism Descriptions: Some sections discussing the mechanisms of UPF's impact on neuropsychiatric health may be difficult to understand for readers unfamiliar with biochemistry and physiology. It would be beneficial to simplify these sections or include visual diagrams to aid comprehension.
- Predominance of Observational Studies: While the literature reviewed is extensive, most of the studies cited are observational, making it challenging to establish causal relationships between UPF consumption and mental health outcomes. These limitations should be more explicitly acknowledged, along with a suggestion for the need for interventional studies.
- Error in the Introduction: There is a typographical error in the subsection discussing highly processed food. This should be corrected to maintain text clarity.
- Visual Representation of Content: Diagrams and illustrations are valuable additions to the manuscript. The second figure, in particular, is a good attempt at presenting multiple aspects of the topic. However, its readability could be improved—consider simplifying it or breaking it down into several smaller graphics.
- Lack of Discussion on Short- and Long-Term Effects of Inflammation: It would be useful to distinguish between the short-term and long-term effects of chronic inflammation to better highlight the health implications of UPF consumption.
- Bibliographic Error in Line 185: Requires correction.
- Section on Gut Microbiota: A few sentences on its composition would be a valuable addition.
- Section Title Should Be “Discussion”: The current title of the third section does not accurately reflect its content. Renaming it to “Discussion” would better align with standard scientific publication practices.
Conclusion:
The manuscript is well-prepared and covers an important topic with significant public health implications. Its strengths include a rich bibliography and a comprehensive discussion of mechanisms and socio-economic context. However, given the breadth of the topic, a more developed discussion section with insightful perspectives on future research directions or preventive strategies would be expected—currently, this section primarily presents dry facts. Nevertheless, the information is specific, well-argued, and relevant.
Final Recommendation: The manuscript is worthy of publication after implementing the suggested revisions, which will enhance its clarity, formal accuracy, and scientific value.
Reviewer Report
Manuscript Title: Possible associations between Ultraprocessed Foods and Neuropsychiatric Outcomes: Putative mechanisms
The manuscript addresses a timely and significant topic concerning the impact of ultra-processed food (UPF) on neuropsychiatric health. Increasingly, research indicates the negative health effects associated with UPF consumption, as well as the benefits of consuming less processed foods. The authors thoroughly discuss potential mechanisms of UPF influence, covering both biological aspects and socio-economic determinants, which adds great value for readers interested in this issue.
The manuscript is based on an extensive literature review, including both epidemiological and experimental studies, which enhances its credibility. The emphasis on prevention and the need for public health policies to reduce UPF consumption is particularly important. However, there are aspects that require refinement to improve the manuscript's clarity and scientific value.
My Comments:
- Complexity of Mechanism Descriptions: Some sections discussing the mechanisms of UPF's impact on neuropsychiatric health may be difficult to understand for readers unfamiliar with biochemistry and physiology. It would be beneficial to simplify these sections or include visual diagrams to aid comprehension.
- Predominance of Observational Studies: While the literature reviewed is extensive, most of the studies cited are observational, making it challenging to establish causal relationships between UPF consumption and mental health outcomes. These limitations should be more explicitly acknowledged, along with a suggestion for the need for interventional studies.
- Error in the Introduction: There is a typographical error in the subsection discussing highly processed food. This should be corrected to maintain text clarity.
- Visual Representation of Content: Diagrams and illustrations are valuable additions to the manuscript. The second figure, in particular, is a good attempt at presenting multiple aspects of the topic. However, its readability could be improved—consider simplifying it or breaking it down into several smaller graphics.
- Lack of Discussion on Short- and Long-Term Effects of Inflammation: It would be useful to distinguish between the short-term and long-term effects of chronic inflammation to better highlight the health implications of UPF consumption.
- Bibliographic Error in Line 185: Requires correction.
- Section on Gut Microbiota: A few sentences on its composition would be a valuable addition.
- Section Title Should Be “Discussion”: The current title of the third section does not accurately reflect its content. Renaming it to “Discussion” would better align with standard scientific publication practices.
Conclusion:
The manuscript is well-prepared and covers an important topic with significant public health implications. Its strengths include a rich bibliography and a comprehensive discussion of mechanisms and socio-economic context. However, given the breadth of the topic, a more developed discussion section with insightful perspectives on future research directions or preventive strategies would be expected—currently, this section primarily presents dry facts. Nevertheless, the information is specific, well-argued, and relevant.
Final Recommendation: The manuscript is worthy of publication after implementing the suggested revisions, which will enhance its clarity, formal accuracy, and scientific value.
Author Response
- Complexity of Mechanism Descriptions: Some sections discussing the mechanisms of UPF's impact on neuropsychiatric health may be difficult to understand for readers unfamiliar with biochemistry and physiology. It would be beneficial to simplify these sections or include visual diagrams to aid comprehension.
- Predominance of Observational Studies: While the literature reviewed is extensive, most of the studies cited are observational, making it challenging to establish causal relationships between UPF consumption and mental health outcomes. These limitations should be more explicitly acknowledged, along with a suggestion for the need for interventional studies.
- Error in the Introduction: There is a typographical error in the subsection discussing highly processed food. This should be corrected to maintain text clarity.
- Visual Representation of Content: Diagrams and illustrations are valuable additions to the manuscript. The second figure, in particular, is a good attempt at presenting multiple aspects of the topic. However, its readability could be improved—consider simplifying it or breaking it down into several smaller graphics.
- Lack of Discussion on Short- and Long-Term Effects of Inflammation: It would be useful to distinguish between the short-term and long-term effects of chronic inflammation to better highlight the health implications of UPF consumption.
- Bibliographic Error in Line 185: Requires correction.
- Section on Gut Microbiota: A few sentences on its composition would be a valuable addition.
- Section Title Should Be “Discussion”: The current title of the third section does not accurately reflect its content. Renaming it to “Discussion” would better align with standard scientific publication practices.
Conclusion: The manuscript is well-prepared and covers an important topic with significant public health implications. Its strengths include a rich bibliography and a comprehensive discussion of mechanisms and socio-economic context. However, given the breadth of the topic, a more developed discussion section with insightful perspectives on future research directions or preventive strategies would be expected—currently, this section primarily presents dry facts. Nevertheless, the information is specific, well-argued, and relevant.
Final Recommendation: The manuscript is worthy of publication after implementing the suggested revisions, which will enhance its clarity, formal accuracy, and scientific value.
Answer: suggestions were considered, and changes are written in red. A new figure and a new table were added, among other changes according to the recommendations. Answer: suggestions were considered, and changes are written in red. A new figure and a new table were added, among other changes according to the recommendations. Please see details in the Cover Letter file.

Round 2
Reviewer 1 Report
Comments and Suggestions for Authors
To the Authors
Following careful revision work by the Authors of revised Ms (Ms ID nutrients-3538782.v2) is improved in terms of readability, and potential impact to the field.